# BACKPROPAGATION THROUGH THE VOID: OPTIMIZING CONTROL VARIATES FOR BLACK-BOX GRADIENT ESTIMATION

**Will Grathwohl, Dami Choi, Yuhuai Wu, Geoffrey Roeder, David Duvenaud**
University of Toronto and Vector Institute
{wgrathwohl, choidami, ywu, roeder, duvenaud}@cs.toronto.edu

## ABSTRACT

Gradient-based optimization is the foundation of deep learning and reinforcement learning, but is difficult to apply when the mechanism being optimized is unknown or not differentiable. We introduce a general framework for learning low-variance, unbiased gradient estimators, applicable to black-box functions of discrete or continuous random variables. Our method uses gradients of a surrogate neural network to construct a control variate, which is optimized jointly with the original parameters. We demonstrate this framework for training discrete latent-variable models. We also give an unbiased, action-conditional extension of the advantage actor-critic reinforcement learning algorithm.

## 1 INTRODUCTION

Gradient-based optimization has been key to most recent advances in machine learning and reinforcement learning. The back-propagation algorithm (Rumelhart & Hinton, 1986), also known as reverse-mode automatic differentiation (Speelpenning, 1980; Rall, 1981) computes exact gradients of deterministic, differentiable objective functions. The reparameterization trick (Williams, 1992; Kingma & Welling, 2014; Rezende et al., 2014) allows backpropagation to give unbiased, low-variance estimates of gradients of expectations of continuous random variables. This has allowed effective stochastic optimization of large probabilistic latent-variable models.

Unfortunately, there are many objective functions relevant to the machine learning community for which backpropagation cannot be applied. In reinforcement learning, for example, the function being optimized is unknown to the agent and is treated as a black box (Schulman et al., 2015a). Similarly, when fitting probabilistic models with discrete latent variables, discrete sampling operations create discontinuities giving the objective function zero gradient with respect to its parameters. Much recent work has been devoted to constructing gradient estimators for these situations. In reinforcement learning, advantage actor-critic methods (Sutton et al., 2000) give unbiased gradient estimates with reduced variance obtained by jointly optimizing the policy parameters with an estimate of the value function. In discrete latent-variable models, low-variance but biased gradient estimates can be given by continuous relaxations of discrete variables (Maddison et al., 2016; Jang et al., 2016).

A recent advance by Tucker et al. (2017) used a continuous relaxation of discrete random variables to build an unbiased and lower-variance gradient estimator, and showed how to tune the free parameters of these relaxations to minimize the estimator's variance during training. We generalize the method of Tucker et al. (2017) to learn a free-form control variate parameterized by a neural network. This gives a lower-variance, unbiased gradient estimator which can be applied to a wider variety of problems. Most notably, our method is applicable even when no continuous relaxation is available, as in reinforcement learning or black-box function optimization.

## 2 BACKGROUND: GRADIENT ESTIMATORS

How can we choose the parameters of a distribution to maximize an expectation? This problem comes up in reinforcement learning, where we must choose the parameters $\theta$ of a policy distribution $\pi(a|s, \theta)$ to maximize the expected reward $\mathbb{E}_{\tau \sim \pi}[R]$ over state-action trajectories $\tau$. It also

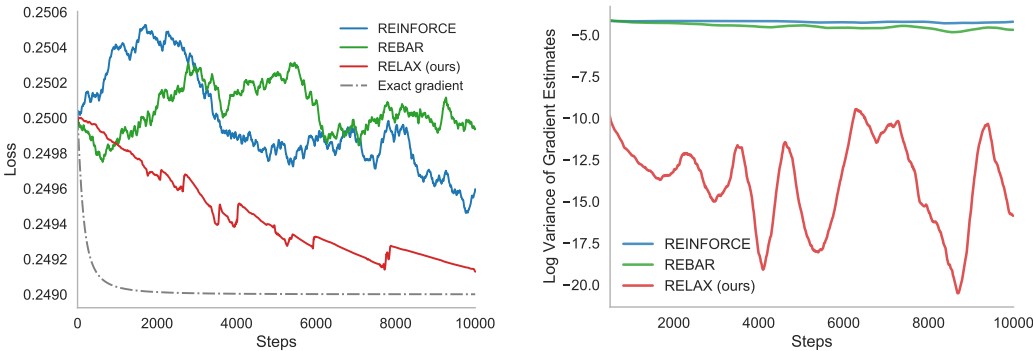

Figure 1: *Left:* Training curves comparing different gradient estimators on a toy problem: $\mathcal{L}(\theta) = \mathbb{E}_{p(b|\theta)}[(b - 0.499)^2]$ *Right:* Log-variance of each estimator's gradient.

comes up in fitting latent-variable models, when we wish to maximize the marginal probability $p(x|\theta) = \sum_z p(x|z)p(z|\theta) = \mathbb{E}_{p(z|\theta)}[p(x|z)]$. In this paper, we'll consider the general problem of optimizing

$$\mathcal{L}(\theta) = \mathbb{E}_{p(b|\theta)}[f(b)]. \tag{1}$$

When the parameters $\theta$ are high-dimensional, gradient-based optimization is appealing because it provides information about how to adjust each parameter individually. Stochastic optimization is essential for scalablility, but is only guaranteed to converge to a fixed point of the objective when the stochastic gradients $\hat{g}$ are unbiased, i.e. $\mathbb{E}[\hat{g}] = \frac{\partial}{\partial \theta}\mathcal{L}(\theta)$ (Robbins & Monro, 1951).

How can we build unbiased, stochastic gradient estimators? There are several standard methods:

**The score-function gradient estimator**   One of the most generally-applicable gradient estimators is known as the score-function estimator, or REINFORCE (Williams, 1992):

$$\hat{g}_{\text{REINFORCE}}[f] = f(b)\frac{\partial}{\partial \theta}\log p(b|\theta), \qquad b \sim p(b|\theta) \tag{2}$$

This estimator is unbiased, but in general has high variance. Intuitively, this estimator is limited by the fact that it doesn't use any information about how $f$ depends on $b$, only on the final outcome $f(b)$.

**The reparameterization trick**   When $f$ is continuous and differentiable, and the latent variables $b$ can be written as a deterministic, differentiable function of a random draw from a fixed distribution, the reparameterization trick (Williams, 1992; Kingma & Welling, 2014; Rezende et al., 2014) creates a low-variance, unbiased gradient estimator by making the dependence of $b$ on $\theta$ explicit through a reparameterization function $b = T(\theta, \epsilon)$:

$$\hat{g}_{\text{reparam}}[f] = \frac{\partial}{\partial \theta}f(b) = \frac{\partial f}{\partial T}\frac{\partial T}{\partial \theta}, \qquad \epsilon \sim p(\epsilon) \tag{3}$$

This gradient estimator is often used when training high-dimensional, continuous latent-variable models, such as variational autoencoders. One intuition for why this gradient estimator is preferable to REINFORCE is that it depends on $\partial f / \partial b$, which exposes the dependence of $f$ on $b$.

**Control variates**   Control variates are a general method for reducing the variance of a stochastic estimator. A control variate is a function $c(b)$ with a known mean $\mathbb{E}_{p(b)}[c(b)]$. Given an estimator $\hat{g}(b)$, subtracting the control variate from this estimator and adding its mean gives us a new estimator:

$$\hat{g}_{\text{new}}(b) = \hat{g}(b) - c(b) + \mathbb{E}_{p(b)}[c(b)] \tag{4}$$

This new estimator has the same expectation as the old one, but has lower variance if $c(b)$ is positively correlated with $\hat{g}(b)$.

# 3 CONSTRUCTING AND OPTIMIZING A DIFFERENTIABLE SURROGATE

In this section, we introduce a gradient estimator for the expectation of a function $\frac{\partial}{\partial \theta} \mathbb{E}_{p(b|\theta)}[f(b)]$ that can be applied even when $f$ is unknown, or not differentiable, or when $b$ is discrete. Our estimator combines the score function estimator, the reparameterization trick, and control variates.

First, we consider the case where $b$ is continuous, but that $f$ cannot be differentiated. Instead of differentiating through $f$, we build a surrogate of $f$ using a neural network $c_\phi$, and differentiate $c_\phi$ instead. Since the score-function estimator and reparameterization estimator have the same expectation, we can simply subtract the score-function estimator for $c_\phi$ and add back its reparameterization estimator. This gives a gradient estimator which we call LAX:

$$\hat{g}_{\text{LAX}} = \hat{g}_{\text{REINFORCE}}[f] - \hat{g}_{\text{REINFORCE}}[c_\phi] + \hat{g}_{\text{reparam}}[c_\phi]$$

$$= [f(b) - c_\phi(b)] \frac{\partial}{\partial \theta} \log p(b|\theta) + \frac{\partial}{\partial \theta} c_\phi(b) \qquad b = T(\theta, \epsilon), \epsilon \sim p(\epsilon). \qquad (5)$$

This estimator is unbiased for any choice of $c_\phi$. When $c_\phi = f$, then LAX becomes the reparameterization estimator for $f$. Thus LAX can have variance at least as low as the reparameterization estimator. An example of the relative bias and variance of each term in this estimator can be seen below.

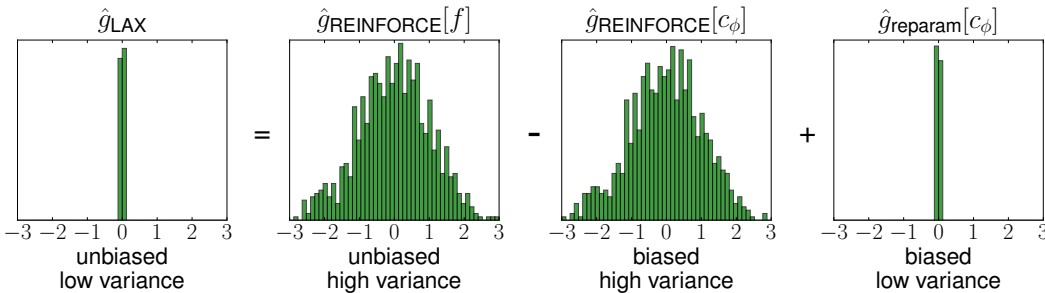

Figure 2: Histograms of samples from the gradient estimators that create LAX. Samples generated from our one-layer VAE experiments (Section 6.2).

## 3.1 GRADIENT-BASED OPTIMIZATION OF THE CONTROL VARIATE

Since $\hat{g}_{\text{LAX}}$ is unbiased for any choice of the surrogate $c_\phi$, the only remaining problem is to choose a $c_\phi$ that gives low variance to $\hat{g}_{\text{LAX}}$. How can we find a $\phi$ which gives our estimator low variance? We simply optimize $c_\phi$ using stochastic gradient descent, at the same time as we optimize the parameters $\theta$ of our model or policy.

To optimize $c_\phi$, we require the gradient of the variance of our estimator. To estimate these gradients, we could simply differentiate through the empirical variance over each mini-batch. Or, following Ruiz et al. (2016a) and Tucker et al. (2017), we can construct an unbiased, single-sample estimator using the fact that our gradient estimator is unbiased. For any unbiased gradient estimator $\hat{g}$ with parameters $\phi$:

$$\frac{\partial}{\partial \phi} \text{Variance}(\hat{g}) = \frac{\partial}{\partial \phi} \mathbb{E}[\hat{g}^2] - \frac{\partial}{\partial \phi} \mathbb{E}[\hat{g}]^2 = \frac{\partial}{\partial \phi} \mathbb{E}[\hat{g}^2] = \mathbb{E}\left[\frac{\partial}{\partial \phi} \hat{g}^2\right]. \qquad (6)$$

Thus, an unbiased single-sample estimate of the gradient of the variance of $\hat{g}$ is given by $\partial \hat{g}^2/\partial \phi$.

This method of directly minimizing the variance of the gradient estimator stands in contrast to other methods such as Q-Prop (Gu et al., 2016) and advantage actor-critic (Sutton et al., 2000), which train the control variate to minimize the squared error $(f(b) - c_\phi(b))^2$. Our algorithm, which jointly optimizes the parameters $\theta$ and the surrogate $c_\phi$ is given in Algorithm 1.

### 3.1.1 OPTIMAL SURROGATE

What is the form of the variance-minimizing $c_\phi$? Inspecting the square of (5), we can see that this loss encourages $c_\phi(b)$ to approximate $f(b)$, but with a weighting based on $\frac{\partial}{\partial \theta} \log p(b|\theta)$. Moreover,

as $c_\phi \to f$ then $\hat{g}_{\text{LAX}} \to \frac{\partial}{\partial\theta}c_\phi$. Thus, this objective encourages a balance between the variance of the reparameterization estimator and the variance of the REINFORCE estimator. Figure 3 shows the learned surrogate on a toy problem.

---

**Algorithm 1** LAX: Optimizing parameters and a gradient control variate simultaneously.

---

**Require:** $f(\cdot), \log p(b|\theta)$, reparameterized sampler $b = T(\theta, \epsilon)$, neural network $c_\phi(\cdot)$,
  step sizes $\alpha_1, \alpha_2$

  **while** not converged **do**
    $\epsilon \sim p(\epsilon)$                                                      ▷ Sample noise
    $b \leftarrow T(\epsilon, \theta)$                                             ▷ Compute input
    $\hat{g}_\theta \leftarrow [f(b) - c_\phi(b)]\, \nabla_\theta \log p(b|\theta) + \nabla_\theta c_\phi(b)$      ▷ Estimate gradient of objective
    $\hat{g}_\phi \leftarrow \partial\hat{g}_\theta^2/\partial\phi$              ▷ Estimate gradient of variance of gradient
    $\theta \leftarrow \theta - \alpha_1\hat{g}_\theta$                            ▷ Update parameters
    $\phi \leftarrow \phi - \alpha_2\hat{g}_\phi$                           ▷ Update control variate
  **end while**
  **return** $\theta$

---

### 3.2 DISCRETE RANDOM VARIABLES AND CONDITIONAL REPARAMETERIZATION

We can adapt the LAX estimator to the case where $b$ is a discrete random variable by introducing a "relaxed" continuous variable $z$. We require a continuous, reparameterizable distribution $p(z|\theta)$ and a deterministic mapping $H(z)$ such that $H(z) = b \sim p(b|\theta)$ when $z \sim p(z|\theta)$. In our implementation, we use the Gumbel-softmax trick, the details of which can be found in appendix B.

The discrete version of the LAX estimator is given by:

$$\hat{g}_{\text{DLAX}} = f(b)\frac{\partial}{\partial\theta}\log p(b|\theta) - c_\phi(z)\frac{\partial}{\partial\theta}\log p(z|\theta) + \frac{\partial}{\partial\theta}c_\phi(z), \qquad b = H(z), z \sim p(z|\theta). \quad (7)$$

This estimator is simple to implement and general. However, if we were able to replace the $\frac{\partial}{\partial\theta}\log p(z|\theta)$ in the control variate with $\frac{\partial}{\partial\theta}\log p(b|\theta)$ we should be able to achieve a more correlated control variate, and therefore a lower variance estimator. This is the motivation behind our next estimator, which we call RELAX.

To construct a more powerful gradient estimator, we incorporate a further refinement due to Tucker et al. (2017). Specifically, we evaluate our control variate both at a relaxed input $z \sim p(z|\theta)$, and also at a relaxed input *conditioned on the discrete variable* $b$, denoted $\tilde{z} \sim p(z|b, \theta)$. Doing so gives us:

$$\hat{g}_{\text{RELAX}} = [f(b) - c_\phi(\tilde{z})]\,\frac{\partial}{\partial\theta}\log p(b|\theta) + \frac{\partial}{\partial\theta}c_\phi(z) - \frac{\partial}{\partial\theta}c_\phi(\tilde{z}) \quad (8)$$
$$b = H(z), z \sim p(z|\theta), \tilde{z} \sim p(z|b, \theta)$$

This estimator is unbiased for any $c_\phi$. A proof and a detailed algorithm can be found in appendix A. We note that the distribution $p(z|b, \theta)$ must also be reparameterizable. We demonstrate how to perform this conditional reparameterization for Bernoulli and categorical random variables in appendix B.

### 3.3 CHOOSING THE CONTROL VARIATE ARCHITECTURE

The variance-reduction objective introduced above allows us to use any differentiable, parametric function as our control variate $c_\phi$. How should we choose the architecture of $c_\phi$? Ideally, we will take advantage of any known structure in $f$.

In the discrete setting, if $f$ is known and happens to be differentiable, we can use the concrete relaxation (Jang et al., 2016; Maddison et al., 2016) and let $c_\phi(z) = f(\sigma_\lambda(z))$. In this special case, our estimator is exactly the REBAR estimator. We are also free to add a learned component to the concrete relaxation and let $c_\phi(z) = f(\sigma_\lambda(z)) + r_\rho(z)$ where $r_\rho$ is a neural network with parameters $\rho$ making $\phi = \{\rho, \lambda\}$. We took this approach in our experiments training discrete variational autoencoders. If $f$ is unknown, we can simply let $c_\phi$ be a generic function approximator such as a neural network. We took this simpler approach in our reinforcement learning experiments.

## 3.4 REINFORCEMENT LEARNING

We now describe how we apply the LAX estimator in the reinforcement learning (RL) setting. By reinforcement learning, we refer to the problem of optimizing the parameters $\theta$ of a policy distribution $\pi(a|s,\theta)$ to maximize the sum of rewards. In this setting, the random variable being integrated over is $\tau$, which denotes a series of $T$ actions and states $[(s_1, a_1), (s_2, a_2), ..., (s_T, a_T)]$. The function whose expectation is being optimized, $R$, maps $\tau$ to the sum of rewards $R(\tau) = \sum_{t=1}^{T} r_t(s_t, a_t)$.

Again, we want to estimate the gradient of an expectation of a black-box function: $\frac{\partial}{\partial \theta} \mathbb{E}_{p(\tau|\theta)}[R(\tau)]$. The *de facto* standard approach is the advantage actor-critic estimator (A2C) (Sutton et al., 2000):

$$\hat{g}_{\text{A2C}} = \sum_{t=1}^{T} \frac{\partial \log \pi(a_t|s_t,\theta)}{\partial \theta} \left[ \sum_{t'=t}^{T} r_{t'} - c_\phi(s_t) \right], \qquad a_t \sim \pi(a_t|s_t,\theta) \tag{9}$$

Where $c_\phi(s_t)$ is an estimate of the state-value function, $c_\phi(s) \approx V^\pi(s) = \mathbb{E}_\tau[R|s_1 = s]$. This estimator is unbiased when $c_\phi$ does not depend on $a_t$. The main limitations of A2C are that $c_\phi$ does not depend on $a_t$, and that it's not obvious how to optimize $c_\phi$. Using the LAX estimator addresses both of these problems.

First, we assume $\pi(a_t|s_t,\theta)$ is reparameterizable, meaning that we can write $a_t = a(\epsilon_t, s_t, \theta)$, where $\epsilon_t$ does not depend on $\theta$. We again introduce a differentiable surrogate $c_\phi(a, s)$. Crucially, this surrogate is a function of the action as well as the state.

The extension of LAX to Markov decision processes is:

$$\hat{g}_{\text{LAX}}^{\text{RL}} = \sum_{t=1}^{T} \frac{\partial \log \pi(a_t|s_t,\theta)}{\partial \theta} \left[ \sum_{t'=t}^{T} r_{t'} - c_\phi(a_t, s_t) \right] + \frac{\partial}{\partial \theta} c_\phi(a_t, s_t), \tag{10}$$

$$a_t = a(\epsilon_t, s_t, \theta) \qquad \epsilon_t \sim p(\epsilon_t).$$

This estimator is unbiased if the true dynamics of the system are Markovian w.r.t. the state $s_t$. When $T = 1$, we recover the special case $\hat{g}_{\text{LAX}}^{\text{RL}} = \hat{g}_{\text{LAX}}$. Comparing $\hat{g}_{\text{LAX}}^{\text{RL}}$ to the standard advantage actor-critic estimator in (9), the main difference is that our baseline $c_\phi(a_t, s_t)$ is action-dependent while still remaining unbiased.

To optimize the parameters $\phi$ of our control variate $c_\phi(a_t, s_t)$, we can again use the single-sample estimator of the gradient of our estimator's variance given in (6). This approach avoids unstable training dynamics, and doesn't require storage and replay of previous rollouts.

Details of this derivation, as well as the discrete and conditionally reparameterized version of this estimator can be found in appendix C.

## 4 SCOPE AND LIMITATIONS

The work most related to ours is the recently-developed REBAR method (Tucker et al., 2017), which greatly inspired our work. The REBAR estimator is a special case of the RELAX estimator, when the surrogate is set to $c_\phi(z) = \eta \cdot f(\texttt{softmax}_\lambda(z))$. The only free parameters of the REBAR estimator are the scaling factor $\eta$, and the temperature $\lambda$, which gives limited scope to optimize the surrogate. REBAR can only be applied when $f$ is known and differentiable. Furthermore, it depends on essentially undefined behavior of the function being optimized, since it evaluates the discrete loss function at continuous inputs.

Because LAX and RELAX can construct a surrogate from scratch, they can be used for optimizing black-box functions, as in reinforcement learning settings where the reward is an unknown function of the environment. LAX and RELAX only require that we can query the function being optimized, and can sample from and differentiate $p(b|\theta)$.

**Direct dependence on parameters** Above, we assumed that the function $f$ being optimized does not depend directly on $\theta$, which is usually the case in black-box optimization settings. However, a dependence on $\theta$ can occur when training probabilistic models, or when we add a regularizer. In both

these settings, if the dependence on $\theta$ is known and differentiable, we can use the fact that

$$\frac{\partial}{\partial \theta} \mathbb{E}_{p(b|\theta)}[f(b,\theta)] = \mathbb{E}_{p(b|\theta)} \left[ \frac{\partial}{\partial \theta} f(b,\theta) + f(b,\theta) \frac{\partial}{\partial \theta} \log p(b|\theta) \right] \tag{11}$$

and simply add $\frac{\partial}{\partial \theta} f(b,\theta)$ to any of the gradient estimators above to recover an unbiased estimator.

## 5 RELATED WORK

Miller et al. (2017) reduce the variance of reparameterization gradients in an orthogonal way to ours by approximating the gradient-generating procedure with a simple model and using that model as a control variate. NVIL (Mnih & Gregor, 2014) and VIMCO (Mnih & Rezende, 2016) provide reduced variance gradient estimation in the special case of discrete latent variable models and discrete latent variable models with Monte Carlo objectives. Salimans et al. (2017) estimate gradients using a form of finite differences, evaluating hundreds of different parameter values in parallel to construct a gradient estimate. In contrast, our method is a single-sample estimator.

Staines & Barber (2012) address the general problem of developing gradient estimators for deterministic black-box functions or discrete optimization. They introduce a sampling distribution, and optimize an objective similar to ours. Wierstra et al. (2014) also introduce a sampling distribution to build a gradient estimator, and consider optimizing the sampling distribution. In the context of general Monte Carlo integration, Oates et al. (2017) introduce a non-parametric control variate that also leverages gradient information to reduce the variance of an estimator.

In parallel to our work, there has been a string of recent developments on action-dependent baselines for policy-gradient methods in reinforcement learning. Such works include Gu et al. (2016) and Gu et al. (2017) which train an action-dependent baseline which incorporates off-policy data. Liu et al. (2017) independently develop a method similar to LAX applied to continuous control. Wu et al. (2018) exploit per-dimension independence of the action distribution in continuous control tasks to produce an action-dependent unbiased baseline.

## 6 APPLICATIONS

We demonstrate the effectiveness of our estimator on a number of challenging optimization problems. Following Tucker et al. (2017) we begin with a simple toy example to illuminate the potential of our method and then continue to the more relevant problems of optimizing binary VAE's and reinforcement learning.

### 6.1 TOY EXPERIMENT

As a simple example, we follow Tucker et al. (2017) in minimizing $\mathbb{E}_{p(b|\theta)}[(b - t)^2]$ as a function of the parameter $\theta$ where $p(b|\theta) =$ Bernoulli$(b|\theta)$. Tucker et al. (2017) set the target $t = .45$. We focus on the more challenging case where $t = .499$. Figures 1a and 1b show the relative performance and gradient log-variance of REINFORCE, REBAR, and RELAX.

Figure 3 plots the learned surrogate $c_\phi$ for a fixed value of $\theta$. We can see that $c_\phi$ is near $f$ for all $z$, keeping the variance of the REINFORCE part of the estimator small. Moreover the deriva-

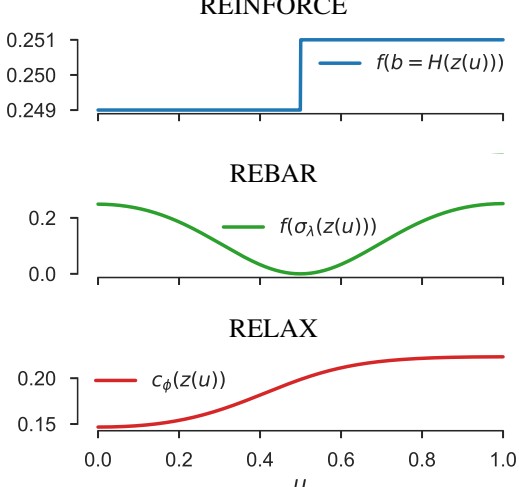

Figure 3: The optimal relaxation for a toy loss function, using different gradient estimators. Because REBAR uses the concrete relaxation of $f$, which happens to be implemented as a quadratic function, the optimal relaxation is constrained to be a warped quadratic. In contrast, RELAX can choose a free-form relaxation.

tive of $c_\phi$ is positive for all $z$ meaning that the reparameterization part of the estimator will produce gradients pointing in the correct direction to optimize the expectation. Conversely, the concrete

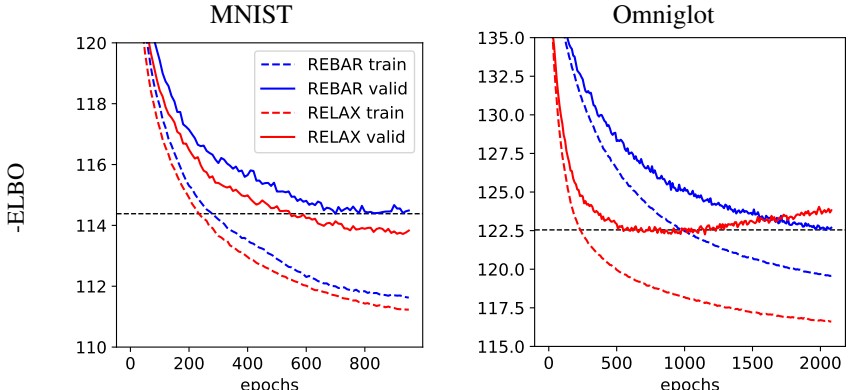

Figure 4: Training curves for the VAE Experiments with the one-layer linear model. The horizontal dashed line indicates the lowest validation error obtained by REBAR.

relaxation of REBAR is close to $f$ only near $0$ and $1$ and its gradient points in the correct direction only for values of $z > \log(\frac{1-t}{t})$. These factors together result in the RELAX estimator achieving the best performance.

## 6.2 DISCRETE VARIATIONAL AUTOENCODER

Next, we evaluate the RELAX estimator on the task of training a variational autoencoder (Kingma & Welling, 2014; Rezende et al., 2014) with Bernoulli latent variables. We reproduced the variational autoencoder experiments from Tucker et al. (2017), training models with one or two layers of 200 Bernoulli random variables with linear or nonlinear mappings between them, on both the MNIST and Omniglot (Lake et al., 2015) datasets. Details of these models and our experimental procedure can be found in Appendix E.1.

To take advantage of the available structure in the loss function, we choose the form of our control variate to be $c_\phi(z) = f(\sigma_\lambda(z)) + \hat{r}_\rho(z)$ where $\hat{r}_\rho$ is a neural network with parameters $\rho$ and $f(\sigma_\lambda(z))$ is the discrete loss function, the evidence lower-bound (ELBO), evaluated at continuously relaxed inputs as in REBAR. In all experiments, the learned control variate improved the training performance, over the state-of-the-art baseline of REBAR. In both linear models, we achieved improved validation performance as well increased convergence speed. We believe the decrease in validation performance for the nonlinear models was due to overfitting caused by improved optimization of an under-regularized model. We leave exploring this phenomenon to further work.

| Dataset | Model | Concrete | NVIL | MuProp | REBAR | RELAX |
|---|---|---|---|---|---|---|
| **MNIST** | Nonlinear | $-102.2$ | $-101.5$ | -101.1 | -81.01 | **-78.13** |
| | linear one-layer | -111.3 | $-112.5$ | $-111.7$ | -111.6 | **-111.20** |
| | linear two-layer | -99.62 | $-99.6$ | $-99.07$ | -98.22 | **-98.00** |
| **Omniglot** | Nonlinear | $-110.4$ | $-109.58$ | -108.72 | -56.76 | **-56.12** |
| | linear one-layer | -117.23 | $-117.44$ | $-117.09$ | -116.63 | **-116.57** |
| | linear two-layer | -109.95 | $-109.98$ | $-109.55$ | -108.71 | **-108.54** |

Table 1: Highest training ELBO for discrete variational autoencoders.

To obtain training curves we created our own implementation of REBAR, which gave identical or slightly improved performance compared to the implementation of Tucker et al. (2017).

While we obtained a modest improvement in training and validation scores (tables 1 and 3), the most notable improvement provided by RELAX is in its rate of convergence. Training curves for all models can be seen in Figure 4 and in Appendix D. In Table 4 we compare the number of training epochs that are required to match the best validation score of REBAR. In both linear models, RELAX provides an increase in rate of convergence.

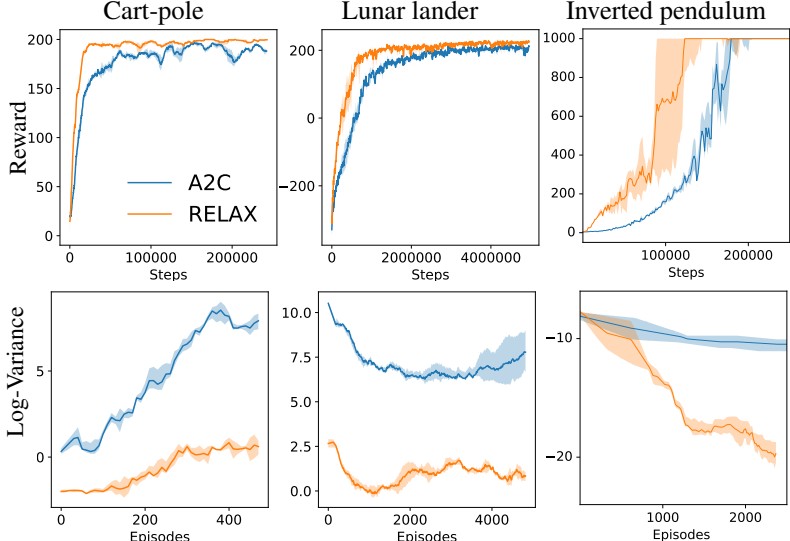

Figure 5: *Top row:* Reward curves. *Bottom row:* Log-variance of policy gradients. In each curve, the center line indicates the mean reward over 5 random seeds. The opaque bars in the top row indicate the 25th and 75th percentiles. The opaque bars in the bottom row indicate 1 standard deviation. Since the gradient estimator is defined at the end of each episode, we display log-variance per episode. After every 10th training episode 100 episodes were run and the sample log-variance is reported averaged over all policy parameters.

| Model | Cart-pole | Lunar lander | Inverted pendulum |
|---|---|---|---|
| A2C | $1152 \pm 90$ | $162374 \pm 17241$ | $6243 \pm 164$ |
| LAX/RELAX | $\mathbf{472 \pm 114}$ | $\mathbf{68712 \pm 20668}$ | $\mathbf{2067 \pm 412}$ |

Table 2: Mean episodes to solve tasks. Definitions of solving each task can be found in Appendix E.

## 6.3 REINFORCEMENT LEARNING

We apply our gradient estimator to a few simple reinforcement learning environments with discrete and continuous actions. We use the RELAX and LAX estimators for discrete and continuous actions, respectively. We compare with the advantage actor-critic algorithm (A2C) (Sutton et al., 2000) as a baseline.

As our control variate does not have the same interpretation as the value function of A2C, it was not directly clear how to add reward bootstrapping and other variance reduction techniques common in RL into our model. For instance, to do reward bootstrapping, we would need to use the state-value function. In the discrete experiments, due to the simplicity of the tasks, we chose not to use reward bootstrapping, and therefore omitted the use of state-value function. However, with the more complicated continuous tasks, we chose to use the value function to enable bootstrapping. In this case, the control variate takes the form: $c_\phi(a, s) = V(s) + \hat{c}(a, s)$, where $V(s)$ is trained as it would be in A2C. Full details of our experiments can be found in Appendix E.

In the discrete action setting, we test our approach on the Cart Pole and Lunar Lander environments as provided by the OpenAI gym (Brockman et al., 2016). In the continuous action setting, we test on the MuJoCo-simulated (Todorov et al., 2012) environment Inverted Pendulum also found in the OpenAI gym. In all tested environments we observe improved performance and sample efficiency using our method. The results of our experiments can be seen in Figure 5, and Table 2.

We found that our estimator produced policy gradients with drastically reduced variance (see Figure 5) allowing for larger learning rates to be used while maintaining stable training. In both discrete environments our estimator achieved greater than a 2-times speedup in convergence over the baseline.

## 7 Conclusions and future work

In this work we synthesized and generalized several standard approaches for constructing gradient estimators. We proposed a generic gradient estimator that can be applied to expectations of known or black-box functions of discrete or continuous random variables, and adds little computational overhead. We also derived a simple extension to reinforcement learning in both discrete and continuous-action domains.

Future applications of this method could include training models with hard attention or memory indexing (Zaremba & Sutskever, 2015). One could also apply our estimators to continuous latent-variable models whose likelihood is non-differentiable, such as a 3D rendering engine. Extensions to the reparameterization gradient estimator (Ruiz et al., 2016b; Naesseth et al., 2017) could also be applied to increase the scope of distributions that can be modeled.

In the reinforcement learning setting, our method could be combined with other variance-reduction techniques such as generalized advantage estimation (Kimura et al., 2000; Schulman et al., 2015b), or other optimization methods, such as KFAC (Wu et al., 2017). One could also train our control variate off-policy, as in $Q$-prop (Gu et al., 2016).

### Acknowledgements

We thank Dougal Maclaurin, Tian Qi Chen, Elliot Creager, and Bowen Xu for helpful discussions. We also thank Christopher Prohm for pointing out an error in one of our derivations. We would also like to thank George Tucker for pointing out a bug in our initially released reinforcement learning code.

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

## APPENDICES

## A    THE RELAX ALGORITHM

*Proof.* We show that $\hat{g}_{\text{RELAX}}$ is an unbiased estimator of $\frac{\partial}{\partial\theta}\mathbb{E}_{p(b|\theta)}[f(b)]$. The estimator is

$$\mathbb{E}_{p(b|\theta)}\left[\left[f(b) - \mathbb{E}_{p(\tilde{z}|b,\theta)}[c_\phi(\tilde{z})]\right]\frac{\partial}{\partial\theta}\log p(b|\theta) - \frac{\partial}{\partial\theta}\mathbb{E}_{p(\tilde{z}|b,\theta)}[c_\phi(\tilde{z})]\right] + \frac{\partial}{\partial\theta}\mathbb{E}_{p(z|\theta)}[c_\phi(z)].$$

Expanding the expectation for clarity of exposition, we account for each term in the estimator separately:

$$\mathbb{E}_{p(b|\theta)}\left[f(b)\frac{\partial}{\partial\theta}\log p(b|\theta)\right] \tag{12}$$

$$-\mathbb{E}_{p(b|\theta)}\left[\mathbb{E}_{p(\tilde{z}|b,\theta)}[c_\phi(\tilde{z})]\frac{\partial}{\partial\theta}\log p(b|\theta)\right] \tag{13}$$

$$-\mathbb{E}_{p(b|\theta)}\left[\frac{\partial}{\partial\theta}\mathbb{E}_{p(\tilde{z}|b,\theta)}[c_\phi(\tilde{z})]\right] \tag{14}$$

$$+\frac{\partial}{\partial\theta}\mathbb{E}_{p(z|\theta)}[c_\phi(z)]. \tag{15}$$

Term (12) is an unbiased score-function estimator of $\frac{\partial}{\partial\theta}\mathbb{E}_{p(b|\theta)}[f(b)]$. It remains to show that the other three terms are zero in expectation. Following Tucker et al. (2017) (see the appendices of that paper for a derivation), we rewrite term (14) as follows:

$$-\mathbb{E}_{p(b|\theta)}\left[\frac{\partial}{\partial\theta}\mathbb{E}_{p(\tilde{z}|b,\theta)}[c_\phi(\tilde{z})]\right] = \mathbb{E}_{p(b|\theta)}\left[\mathbb{E}_{p(\tilde{z}|b,\theta)}[c_\phi(\tilde{z})]\frac{\partial}{\partial\theta}\log p(b|\theta)\right]$$
$$-\mathbb{E}_{p(z|\theta)}\left[c_\phi(z)\frac{\partial}{\partial\theta}\log p(z)\right]. \tag{16}$$

Note that the first term on the right-hand side of equation (16) is equal to term (13) with opposite sign. The second term on the right-hand side of equation (16) is the score-function estimator of term (15), opposite in sign. The sum of these terms is zero in expectation.

$\square$

---

**Algorithm 2** RELAX: Low-variance control variate optimization for black-box gradient estimation.

**Require:** $f(\cdot)$, $\log p(b|\theta)$, reparameterized samplers $b = H(z)$, $z = S(\epsilon, \theta)$ and $\tilde{z} = S(\epsilon, \theta|b)$,
          neural network $c_\phi(\cdot)$, step sizes $\alpha_1, \alpha_2$

  **while** not converged **do**
    $\epsilon_i, \widetilde{\epsilon}_i \sim p(\epsilon)$                                            ▷ Sample noise
    $z_i \leftarrow S(\epsilon_i, \theta)$                        ▷ Compute unconditional relaxed input
    $b_i \leftarrow H(z_i)$                                 ▷ Compute input
    $\widetilde{z}_i \leftarrow S(\widetilde{\epsilon}_i, \theta|b_i)$                    ▷ Compute conditional relaxed input
    $\hat{g}_\theta \leftarrow [f(b_i) - c_\phi(\widetilde{z}_i)]\nabla_\theta \log p + \nabla_\theta c_\phi(z_i) - \nabla_\theta c_\phi(\widetilde{z}_i)$   ▷ Estimate gradient
    $\hat{g}_\phi \leftarrow \partial\hat{g}_\theta^2/\partial\phi$                  ▷ Estimate gradient of variance of gradient
    $\theta \leftarrow \theta - \alpha_1\hat{g}_\theta$                            ▷ Update parameters
    $\phi \leftarrow \phi - \alpha_2\hat{g}_\phi$                         ▷ Update control variate
  **end while**
  **return** $\theta$

---

## B    CONDITIONAL RE-SAMPLING FOR DISCRETE RANDOM VARIABLES

When applying the RELAX estimator to a function of discrete random variables $b \sim p(b|\theta)$, we require that there exists a distribution $p(z|\theta)$ and a deterministic mapping $H(z)$ such that if $z \sim p(z|\theta)$

then $H(z) = b \sim p(b|\theta)$. Treating both $b$ and $z$ as random, this procedure defines a probabilistic model $p(b, z|\theta) = p(b|z)p(z|\theta)$. The RELAX estimator requires reparameterized samples from $p(z|\theta)$ and $p(z|b, \theta)$. We describe how to sample from these distributions in the common cases of $p(b|\theta) = \text{Bernoulli}(\theta)$ and $p(b|\theta) = \text{Categorical}(\theta)$.

**Bernoulli**   When $p(b|\theta)$ is Bernoulli distribution we let $H(z) = \mathbb{I}(z > 0)$ and we sample from $p(z|\theta)$ with

$$z = \log \frac{\theta}{1 - \theta} + \log \frac{u}{1 - u}, \qquad u \sim \text{uniform}[0, 1].$$

We can sample from $p(z|b, \theta)$ with

$$v' = \begin{cases} v \cdot (1 - \theta) & b = 0 \\ v \cdot \theta + (1 - \theta) & b = 1 \end{cases}$$

$$\tilde{z} = \log \frac{\theta}{1 - \theta} + \log \frac{v'}{1 - v'}, \qquad v \sim \text{uniform}[0, 1].$$

**Categorical**   When $p(b|\theta)$ is a Categorical distribution where $\theta_i = p(b = i|\theta)$, we let $H(z) = \text{argmax}(z)$ and we sample from $p(z|\theta)$ with

$$z = \log \theta - \log(-\log u), \qquad u \sim \text{uniform}[0, 1]^k$$

where $k$ is the number of possible outcomes.

To sample from $p(z|b, \theta)$, we note that the distribution of the largest $\hat{z}_b$ is independent of $\theta$, and can be sampled as $\hat{z}_b = -\log(-\log v_b)$ where $v_b \sim \text{uniform}[0, 1]$. Then, the remaining $v_{i \neq b}$ can be sampled as before but with their underlying noise truncated so $\hat{z}_{i \neq b} < \hat{z}_b$. As shown in the appendix of Tucker et al. (2017), we can then sample from $p(z|b, \theta)$ with:

$$\hat{z}_i = \begin{cases} -\log(-\log v_i) & i = b \\ -\log\left(-\frac{\log v_i}{\theta_i} - \log v_b\right) & i \neq b \end{cases} \tag{17}$$

where $v_i \sim \text{uniform}[0, 1]$.

## C   DERIVATIONS OF ESTIMATORS USED IN REINFORCEMENT LEARNING

We give the derivation of the LAX estimator used for continuous RL tasks.

**Theorem C.1.** *The* LAX *estimator,*

$$\hat{g}_{\text{LAX}}^{\text{RL}} = \sum_{t=1}^{T} \frac{\partial \log \pi(a_t|s_t, \theta)}{\partial \theta} \left[ \sum_{t'=t}^{T} r_{t'} - c_\phi(a_t, s_t) \right] + \frac{\partial}{\partial \theta} c_\phi(a_t, s_t), \tag{18}$$

$$a_t = a_t(\epsilon_t, s_t, \theta), \quad \epsilon_t \sim p(\epsilon_t),$$

*is unbiased.*

*Proof.* Note that by using the score-function estimator, for all $t$, we have

$$\mathbb{E}_{p(\tau)}\left[ \frac{\partial \log \pi(a_t|s_t, \theta)}{\partial \theta} c_\phi(a_t, s_t) \right] = \mathbb{E}_{p(a_{1:t-1}, s_{1:t})}\left[ \frac{\partial}{\partial \theta} \mathbb{E}_{\pi(a_t|s_t, \theta)}\left[ c_\phi(a_t, s_t) \right] \right].$$

Then, by adding and subtracting the same term, we have

$$
\frac{\partial}{\partial \theta} \mathbb{E}_{p(\tau)}[f(\tau)] = \mathbb{E}_{p(\tau)} \left[ f(\tau) \cdot \frac{\partial}{\partial \theta} \log p(\tau; \theta) \right] - \sum_t \mathbb{E}_{p(\tau)} \left[ \frac{\partial \log \pi(a_t|s_t, \theta)}{\partial \theta} c_\phi(a_t, s_t) \right] +
$$

$$
\sum_t \mathbb{E}_{p(a_{1:t-1}, s_{1:t})} \left[ \frac{\partial}{\partial \theta} \mathbb{E}_{\pi(a_t|s_t, \theta)} \left[ c_\phi(a_t, s_t) \right] \right]
$$

$$
= \mathbb{E}_{p(\tau)} \left[ \sum_{t=1}^{\infty} \frac{\partial \log \pi(a_t|s_t, \theta)}{\partial \theta} \left( \sum_{t'=t}^{\infty} r_{t'} - c_\phi(a_t, s_t) \right) \right]
$$

$$
+ \sum_t \mathbb{E}_{p(a_{1:t-1}, s_{1:t})} \left[ \mathbb{E}_{p(\epsilon_t)} \left[ \frac{\partial}{\partial \theta} c_\phi(a_t(\epsilon_t, s_t, \theta), s_t) \right] \right]
$$

$$
= \mathbb{E}_{p(\tau)} \left[ \sum_{t=1}^{\infty} \frac{\partial \log \pi(a_t|s_t, \theta)}{\partial \theta} \left( \sum_{t'=t}^{\infty} r_{t'} - c_\phi(a_t, s_t) \right) + \frac{\partial}{\partial \theta} c_\phi(a_t(\epsilon_t, s_t, \theta), s_t) \right]
$$

$$\square$$

In the discrete control setting, our policy parameterizes a soft-max distribution which we use to sample actions. We define $z_t \sim p(z_t|s_t)$, which is equal to $\sigma(\log \pi - \log(-\log(u)))$ where $u \sim$ uniform$[0, 1]$, $a_t = \text{argmax}(z_t)$, $\sigma$ is the soft-max function. We also define $\tilde{z}_t \sim p(z_t|a_t, s_t)$ and uses the same reparametrization trick for sampling $\tilde{z}_t$ as explicated in Appendix B.

**Theorem C.2.** *The* RELAX *estimator,*

$$
\hat{g}_{\text{RELAX}}^{\text{RL}} = \sum_{t=1}^{T} \frac{\partial \log \pi(a_t|s_t, \theta)}{\partial \theta} \left( \sum_{t'=t}^{T} r_{t'} - c_\phi(\tilde{z}_t, s_t) \right) - \frac{\partial}{\partial \theta} c_\phi(\tilde{z}_t, s_t) + \frac{\partial}{\partial \theta} c_\phi(z_t, s_t), \quad (19)
$$

$$
\tilde{z}_t \sim p(z_t|a_t, s_t), \qquad z_t \sim p(z_t|s_t),
$$

*is unbiased.*

*Proof.* Note that by using the score-function estimator, for all $t$, we have

$$
\mathbb{E}_{p(a_{1:t}, s_{1:t})} \left[ \frac{\partial \log \pi(a_t|s_t, \theta)}{\partial \theta} \mathbb{E}_{p(z_t|a_t, s_t)}[c_\phi(z_t, s_t)] \right]
$$

$$
= \mathbb{E}_{p(a_{1:t-1}, s_{1:t})} \left[ \frac{\partial}{\partial \theta} \mathbb{E}_{\pi(a_t|s_t, \theta)} \left[ \mathbb{E}_{p(z_t|a_t, s_t)}[c_\phi(z_t, s_t)] \right] \right]
$$

$$
= \mathbb{E}_{p(a_{1:t-1}, s_{1:t})} \left[ \frac{\partial}{\partial \theta} \mathbb{E}_{p(z_t|s_t)}[c_\phi(z_t, s_t)] \right]
$$

Then, by adding and subtracting the same term, we have

$$
\frac{\partial}{\partial \theta} \mathbb{E}_{p(\tau)}[f(\tau)] = \mathbb{E}_{p(\tau)} \left[ f(\tau) \cdot \frac{\partial}{\partial \theta} \log p(\tau; \theta) \right]
$$

$$
- \sum_t \mathbb{E}_{p(a_{1:t}, s_{1:t})} \left[ \frac{\partial \log \pi(a_t|s_t, \theta)}{\partial \theta} \mathbb{E}_{p(z_t|a_t, s_t)}[c_\phi(z_t, s_t)] \right]
$$

$$
+ \sum_t \mathbb{E}_{p(a_{1:t-1}, s_{1:t})} \left[ \frac{\partial}{\partial \theta} \mathbb{E}_{p(z_t|s_t)}[c_\phi(z_t, s_t)] \right]
$$

$$
= \mathbb{E}_{p(\tau)} \left[ \sum_{t=1}^{\infty} \frac{\partial \log \pi(a_t|s_t, \theta)}{\partial \theta} \left( \sum_{t'=t}^{\infty} r_{t'} - \mathbb{E}_{p(z_t|a_t, s_t)}[c_\phi(z_t, s_t)] \right) \right]
$$

$$
+ \sum_t \mathbb{E}_{p(a_{1:t-1}, s_{1:t})} \left[ \frac{\partial}{\partial \theta} \mathbb{E}_{p(z_t|s_t)}[c_\phi(z_t, s_t)] \right]
$$

$$
= \mathbb{E}_{p(\tau)} \left[ \sum_{t=1}^{\infty} \frac{\partial \log \pi(a_t|s_t, \theta)}{\partial \theta} \left( \sum_{t'=t}^{\infty} r_{t'} - \mathbb{E}_{p(z_t|a_t, s_t)}[c_\phi(z_t, s_t)] \right) \right.
$$

$$
\left. - \frac{\partial}{\partial \theta} \mathbb{E}_{p(z_t|a_t, s_t)}[c_\phi(z_t, s_t)] + \frac{\partial}{\partial \theta} \mathbb{E}_{p(z_t|s_t)}[c_\phi(z_t, s_t)] \right]
$$

Since $p(z_t|s_t)$ is reparametrizable, we obtain the estimator in Eq.(19). $\square$

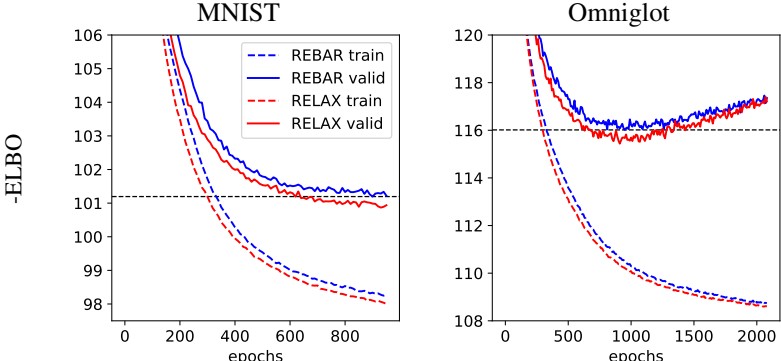

Figure 6: Training curves for the VAE Experiments with the two-layer linear model. The horizontal dashed line indicates the lowest validation error obtained by REBAR.

## D    FURTHER RESULTS ON DISCRETE VARIATIONAL AUTOENCODERS

| Dataset | Model | REBAR | RELAX |
|---------|-------|-------|-------|
| **MNIST** | one-layer linear | -114.32 | **-113.62** |
|  | two-layer linear | -101.20 | **-100.85** |
|  | Nonlinear | **-111.12** | 119.19 |
| **Omniglot** | one-layer linear | -122.44 | **-122.11** |
|  | two-layer linear | -115.83 | **-115.42** |
|  | Nonlinear | **-127.51** | 128.20 |

Table 3: Highest obtained validation ELBO.

| Dataset | Model | REBAR | RELAX |
|---------|-------|-------|-------|
| **MNIST** | one-layer | 857 | **531** |
|  | two-layer | 900 | **620** |
|  | Nonlinear | **331** | - |
| **Omniglot** | one-layer | 2086 | **566** |
|  | two-layer | 1027 | **673** |
|  | Nonlinear | **368** | - |

Table 4: Epochs needed to achieve REBAR's best validation score. "-" indicates that the nonlinear RELAX models achieved lower validation scores than REBAR.

## E    EXPERIMENTAL DETAILS

### E.1    DISCRETE VAE

We run all models for $2,000,000$ iterations with a batch size of $24$. For the REBAR models, we tested learning rates in $\{.005, .001, .0005, .0001, .00005\}$.

RELAX adds more hyperparameters. These are the depth of the neural network component of our control variate $r_\rho$, the weight decay placed on the network, and the scaling on the learning rate for the control variate. We tested neural network models with $l$ layers of 200 units using the ReLU nonlinearity with $l \in \{2, 4\}$. We trained the control variate with weight decay in $\{.001, .0001\}$. We trained the control variate with learning rate scaling in $\{1, 10\}$.

To limit the size of hyperparameter search for the RELAX models, we only test the best performing learning rate for the REBAR baseline and the next largest learning rate in our search set. In many cases, we found that RELAX allowed our model to converge at learning rates which made the REBAR

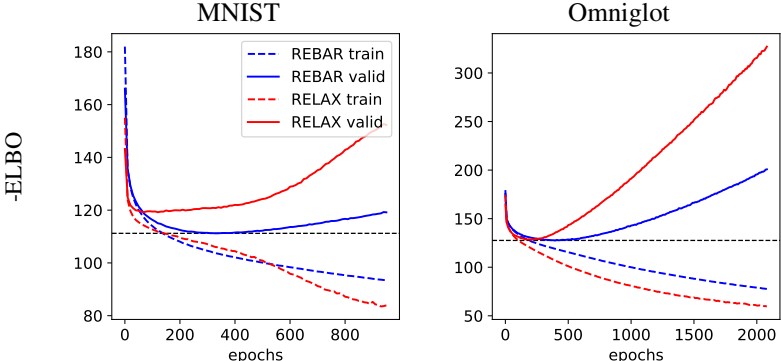

Figure 7: Training curves for the VAE Experiments with the one-layer nonlinear model. The horizontal dashed line indicates the lowest validation error obtained by REBAR.

estimators diverge. We believe further improvement could be achieved by tuning this parameter. It should be noted that in our experiments, we found the RELAX method to be fairly insensitive to all hyperparameters other than learning rate. In general, we found the larger (4 layer) control variate architecture with weight decay of .001 and learning rate scaling of 1 to work best, but only slightly outperformed other configurations.

All presented results are from the models which achieve the highest ELBO on the validation data.

### E.1.1 ONE-LAYER LINEAR MODEL

In the one-layer linear models we optimize the evidence lower bound (ELBO):

$$\log p(x) \geq \mathcal{L}(\theta) = \mathbb{E}_{q(b|x)}[\log p(x|b) + \log p(b) - \log q(b|x)]$$

where $q(b_1|x) = \sigma(x \cdot W_q + \beta_q)$ and $p(x|b_1) = \sigma(b_1 \cdot W_p + \beta_p)$ with weight matrices $W_q, W_p$ and bias vectors $\beta_q, \beta_p$. The parameters of the prior $p(b)$ are also learned.

### E.1.2 TWO LAYER LINEAR MODEL

In the two layer linear models we optimize the ELBO

$$\mathcal{L}(\theta) = \mathbb{E}_{q(b_2|b_1)q(b_1|x)}[\log p(x|b_1) + \log p(b_1|b_2) + \log p(b_2) - \log q(b_1|x) - \log q(b_2|b_1)]$$

where $q(b_1|x) = \sigma(x \cdot W_{q_1} + \beta_{q_1})$, $q(b_2|b_1) = \sigma(b_1 \cdot W_{q_2} + \beta_{q_2})$, $p(x|b_1) = \sigma(b_1 \cdot W_{p_1} + \beta_{p_1})$, and $p(b_1|b_2) = \sigma(b_2 \cdot W_{p_2} + \beta_{p_2})$ with weight matrices $W_{q_1}, W_{q_2}, W_{p_1}, W_{p_2}$ and biases $\beta_{q_1}, \beta_{q_2}, \beta_{p_1}, \beta_{p_2}$. As in the one-layer model, the prior $p(b_2)$ is also learned.

### E.1.3 NONLINEAR MODEL

In the one-layer nonlinear model, the mappings between random variables consist of 2 deterministic layers with 200 units using the hyperbolic-tangent nonlinearity followed by a linear layer with 200 units.

We run an identical hyperpameter search in all models.

### E.2 DISCRETE RL

In both the baseline A2C and RELAX models, the policy and control variate (value function in the baseline model) were two-layer neural networks with 10 units per layer. The ReLU non linearity was used on all layers except for the output layer which was linear.

For these tasks we estimate the policy gradient with a single Monte Carlo sample. We run one episode of the environment to completion, compute the discounted rewards, and run one iteration of gradient descent. We believe using larger batches will improve performance but would less clearly demonstrate the potential of our method.

Both models were trained with the RMSProp (Tieleman & Hinton, 2012) optimizer and a reward discount factor of .99 was used. Entropy regularization with a weight of .01 was used to encourage exploration.

Both models have 2 hyperparameters to tune; the global learning rate and the scaling factor on the learning rate for the control variate (or value function). We complete a grid search for both parameters in $\{0.01, 0.003, 0.001\}$ and present the model which "solves" the task in the fewest number of episodes averaged over 5 random seeds. "Solving" the tasks was defined by the creators of the OpenAI gym (Brockman et al., 2016). The Cart Pole task is considered solved if the agent receives an average reward greater than 195 over 100 consecutive episodes. The Lunar Lander task is considered solved if the agent receives an average reward greater than 200 over 100 consecutive episodes.

The Cart Pole experiments were run for 250,000 frames. The Lunar Lander experiments were run for 5,000,000 frames.

The results presented for the CartPole and LunarLander environments were obtained using a slightly biased sampler for $p(z|b, \theta)$.

### E.3   CONTINUOUS RL

The three models- policy, value, and control variate, are two-layer neural networks with 64 hidden units per layer. The value and control variate networks are identical, with the ELU (Djork-Arné Clevert & Hochreiter, 2016) nonlinearity in each hidden layer. The policy network has `tanh` nonlinearity. The policy network, which parameterizes the Gaussian policy comprises of a network (with the architecture mentioned above) that outputs the mean, and a separate, trainable log standard deviation value that is not input dependent. All three networks have a linear output layer. We selected the batch size to be 2500, meaning for a fixed timestep (2500) we collect multiple rollouts of a task and update the networks' parameters with the batch of episodes. Per one policy update, we optimize both the value and control variate network multiple times. The number of times we train the value network is fixed to 25, while for the control variate, it was chosen to be a hyperparameter. All models were trained using ADAM (Kingma & Ba, 2015), with $\beta_1 = 0.9$, $\beta_2 = 0.999$, and $\epsilon = 1e - 08$.

The baseline A2C case has 2 hyperparameters to tune: the learning rate for the optimizer for the policy and value network. A grid search was done over the set: $\{0.03, 0.003, 0.0003\}$. RELAX has 4 hyperparameters to tune: 3 learning rates for the optimizer per network, and the number of training iterations of the control variate per policy gradient update. Due to the large number of hyperparameters, we restricted the size of the grid search set to $\{0.003, 0.0003\}$ for the learning rates, and $\{1, 5, 25\}$ for the control variate training iteration number. We chose the hyperparameter setting that yielded the shortest episode-to-completion time averaged over 5 random seeds. As with the discrete case, we used the definition of completion provided by the OpenAI gym (Brockman et al., 2016) for each task.

The Inverted Pendulum experiments were run for 1,000,000 frames.

Tucker et al. (2018) pointed out a bug in our initially released code for the continuous RL experiments. This issue has been fixed in the publicly available code and the results presented in this paper were generated with the corrected code.

### E.3.1   IMPLEMENTATION CONSIDERATIONS

For continuous RL tasks, it is convention to employ a batch of a fixed number of timesteps (here, 2500) in which the number of episodes vary. We follow this convention for the sake of providing a fair comparison to the baseline. However, this causes a complication when calculating the variance loss for the control variate because we must compute the variance averaged over completed episodes, which is difficult to obtain when the number of episodes is not fixed. For this reason, in our implementation we compute the gradients for the control variate outside of the Tensorflow computation graph. However, for practical reasons we recommend using a batch of fixed number of episodes when using our method.

