# OpenReview forum: "Backpropagation through the Void: Optimizing control variates for black-box gradient estimation"
_ICLR.cc/2018/Conference — Accept (Poster)_

### Official Review · AnonReviewer2 · 2017-11-11
**Good paper, convincing experiments, writing is clear but there are quite a few typos**

**Rating:** 8
**Confidence:** 4

**Review:**

This paper introduces LAX/RELAX, a method to reduce the variance of the REINFORCE gradient estimator. The method builds on and is directly inspired by REBAR. Similarly to REBAR, RELAX is an unbiased estimator, and the idea is to introduce a control variate that leverages the reparameterization gradient. In contrast to REBAR, RELAX learns a free-from control variate, which allows for low-variance gradient estimates for both discrete and continuous random variables. The method is evaluated on a toy experiment, as well as the discrete VAE and reinforcement learning. It effectively reduces the variance of state-of-the-art methods (namely, REBAR and actor-critic).

Overall, I enjoyed reading the paper. I think it is a neat idea that can be of interest for researchers in the field. The paper is clearly explained, and I found the experiments convincing. I have minor comments only.

+ Is there a good way to initialize c_phi prior to optimization? Given that c_phi must be a proxy for f(), maybe you can take advantage of this observation to find a good initialization for phi?

+ I was confused with the Bernoulli example in Appendix B. Consider the case theta=0.5. Then, b=H(z) takes value 1 if z>0, and 0 otherwise. Thus, p(z|b,theta) should assign mass zero to values z>0 when b=0, which does not seem to be the case with the proposed sampling scheme in page 11, since v*theta=0.5*v, which gives values in [0,0.5]. And similarly for the case b=1.

+ Why is the method called LAX? What does it stand for?

+ In Section 3.3, it is unclear to me why rho!=phi. Given that c_phi(z)=f(sigma_lambda(z))+r_rho(z), with lambda being a temperature parameter, why isn't rho renamed as phi? (the first term doesn't seem to have any parameters). In general, this section was a little bit unclear if you are not familiar with the REBAR method; consider adding more details.

+ Consider adding a brief review of the REBAR estimator in the Background section for those readers who are less familiar with this approach.

+ In the abstract, consider adding two of the main ideas that the estimator relies on: control variates and reparameterization gradients. This would probably be more clear than "based on gradients of a learned function."

+ In the first paragraph of Section 3, the sentence "f is not differentiable or not computable" may be misleading, because it is unclear what "not computable" means (one may think that it cannot be evaluated). Consider replacing with "not analytically computable."

+ In Section 3.3, it reads "differentiable function of discrete random variables," which does not make sense.

+ Before Eq. 11, it reads "where epsilon_t does not depend on theta". I think it should be the distribution over epsilon_t what doesn't depend on theta.

+ In Section 6.1, it was unclear to me why t=.499 is a more challenging setting.

+ The header of Section 6.3.1 should be removed, as Section 6.3 is short.

+ In Section 6.3.1, there is a broken reference to a figure.

+ Please avoid contractions (doesn't, we'll, it's, etc.)

+ There were some other typos; please read carefully the paper and double-check the writing. In particular, I found some missing commas, some proper nouns that are not capitalized in Section 5, and others (e.g., "an learned," "gradient decent").

---

> ### Author Response · Authors · 2017-12-31
> **Response to reviewer 2**
>
> Dear reviewer 2,
>
> Thank you for your kind words and detailed feedback.  Some small changes have been made to address your comments. I will address your comments in the order they were written. Your original comments will appear in [brackets] and my responses will follow.
>
> [+ Is there a good way to initialize c_phi prior to optimization? Given that c_phi must be a proxy for f(), maybe you can take advantage of this observation to find a good initialization for phi?]
>
> Good question. This was touched upon in section 3.3 where we use the concrete relaxation to initialize our control variate and then learn an offset parameterized by a neural network. However this approach is only applicable when f is known.  We also did not experiment with different structures for the control variate beyond different neural network architectures. This is an interesting idea which we leave to further work to explore.
>
>
> [+ I was confused with the Bernoulli example in Appendix B. Consider the case theta=0.5. Then, b=H(z) takes value 1 if z>0, and 0 otherwise. Thus, p(z|b,theta) should assign mass zero to values z>0 when b=0, which does not seem to be the case with the proposed sampling scheme in page 11, since v*theta=0.5*v, which gives values in [0,0.5]. And similarly for the case b=1.]
>
> This issue was due to a typo in Appendix B where theta was swapped for (1 - theta). This has been fixed in the new version.
>
>
> [+ Why is the method called LAX? What does it stand for?]
>
> We first coined “RELAX” as an alternative to REBAR that learned the continuous “relax”-ation.  We then developed LAX, and since it was a simpler version of relax, we chose a simpler name.  We realize that these aren’t particularly descriptive names, and welcome any suggestions for naming these estimators.
>
> [+ In Section 3.3, it is unclear to me why rho!=phi. Given that c_phi(z)=f(sigma_lambda(z))+r_rho(z), with lambda being a temperature parameter, why isn't rho renamed as phi? (the first term doesn't seem to have any parameters). In general, this section was a little bit unclear if you are not familiar with the REBAR method; consider adding more details.]
>
> The paper has been updated to specify that phi = {rho, lambda}
>
>
> [+ Consider adding a brief review of the REBAR estimator in the Background section for those readers who are less familiar with this approach.]
>
> The paper mentions that REBAR can be viewed as a special case of the RELAX method where the concrete relaxation is used as the control variate. For brevity’s sake, a full explanation of REBAR was left out.
>
>
> [+ In the abstract, consider adding two of the main ideas that the estimator relies on: control variates and reparameterization gradients. This would probably be more clear than "based on gradients of a learned function."]
>
> The abstract has been slightly changed to mention that the method involves control variates.
>
>
> [+ In the first paragraph of Section 3, the sentence "f is not differentiable or not computable" may be misleading, because it is unclear what "not computable" means (one may think that it cannot be evaluated). Consider replacing with "not analytically computable."]
>
> Good point. ”not computable” has been removed from that sentence.
>
>
> [+ In Section 3.3, it reads "differentiable function of discrete random variables," which does not make sense.]
>
> By this, it was meant that the function f is differentiable, but we may be evaluating it only on a discrete input. An example of such a function would be f(x) = x^2, where x = {0, 1}. Here f is differentiable when its domain is the real numbers but we are evaluating it restricted to {0, 1}. This wording was removed to avoid confusion.
>
>
> [+ Before Eq. 11, it reads "where epsilon_t does not depend on theta". I think it should be the distribution over epsilon_t what doesn't depend on theta.]
>
> This section has been reworded to make this more clear
>
> [+ In Section 6.1, it was unclear to me why t=.499 is a more challenging setting.]
>
> The closer that t gets to .5 means that the values of f(0) and f(1) get closer together. This means a Monte Carlo estimator of the gradient will require more samples to converge to the correct value.
>
>
> [+ The header of Section 6.3.1 should be removed, as Section 6.3 is short.]
>
> Section 6.3.1 has been removed and consolidated into 6.3
>
>
> [+ In Section 6.3.1, there is a broken reference to a figure]
>
> Good catch.  The broken reference has been fixed in section 6.3
>
> [+ Please avoid contractions (doesn't, we'll, it's, etc.)]
>
> Contractions have been removed.
>
> [+ There were some other typos; please read carefully the paper and double-check the writing. In particular, I found some missing commas, some proper nouns that are not capitalized in Section 5, and others (e.g., "an learned," "gradient decent").]
>
> These typos have been fixed.  Thank you for your attention to detail, it has improved the text considerably.

---

### Official Review · AnonReviewer1 · 2017-11-22
**A good conceptual contribution with a clear background, a not so good empirical study, but I like it much**

**Rating:** 7
**Confidence:** 3

**Review:**

This paper suggests a new approach to performing gradient descent for blackbox optimization or training discrete latent variable models. The paper gives a very clear account of existing gradient estimators and finds a way to combine them so as to construct and optimize a differentiable surrogate function. The resulting new gradient estimator is then studied both theoretically and empirically. The empirical study shows the benefits of the new estimator for training discrete variational autoencoders and for performing deep reinforcement learning.

To me, the main strengths of the paper is the very clear account of existing gradient estimators (among other things it helped me understand obscurities of the Q-prop paper) and a nice conceptual idea. The empirical study itself is more limited and the paper suffers from a few mistakes and missing information, but to me the good points are enough to warrant publication of the paper in a good conference like ICLR.

Below are my comments for the authors.

---------------------------------
General, conceptual comments:

When reading (6), it is clear that the framework performs regression of $c_\phi$ towards the unknown $f$ simultaneously with optimization over $c_\phi$.
Taking this perspective, I would be glad to see how the regression part performs with respect to standard least square regression,
i.e. just using $||f(b)-c_\phi(b)||^2$ as loss function. You may compare the speed of convergence of $c_\phi$ towards $f$ using (6) and the least squared error.
You may also investigate the role of this regression part into the global g_LAX optimization by studying the evolution of the components of (6).

Related to the above comment, in Algo. 1, you mention "f(.)" as given to the algo. Actually, the algo does not know f itself, otherwise it would not be blackbox optimization. So you may mean different things. In a batch setting, you may give a batch of [x,f(x) (,cost(x)?)] points to the algo. You more probably mean here that you have an "oracle" that, given some x, tells you f(x) on demand. But the way you are sampling x is not specified clearly.

This becomes more striking when you move to reinforcement learning problems, which is my main interest. The RL algorithm itself is not much specified. Does it use a replay buffer (probably not)? Is it on-policy or off-policy (probably on-policy)? What about the exploration policy? I want to know more... Probably you just replace (10) with (11) in A2C, but this is not clearly specified.

In Section 4, can you explain why, in the RL case, you must introduce stochasticity to the inputs? Is this related to the exploration issue (see above)?

Last sentence of conclusion: you are too allusive about the relationship between your learned control variate and the Q-function. I don't get it, and I want to know more...

-----------------------------------
Local comments:

Backpropagation through the void: I don't understand why this title. I'm not a native english speaker, I'm probably missing a reference to something, I would be glad to get it.

Figure 1 right. Caption states variance, but it is log variance. Why does it oscillate so much with RELAX?

Beginning of 3.1: you may state more clearly that optimizing $c_\phi$ the way you do it will also "minimize" the variance, and explain better why ("we require the gradient of the variance of our gradient estimator"...). It took me a while to get it.

In 3.1.1 a weighting based on $\d/\d\theta log p(b)$ => shouldn't you write $... log p(b|\theta)$ as before?

Figure 2 is mentioned in p.3, it should appear much sooner than p6.

In Figure 2, there is nothing about the REINFORCE PART. Why?

In 3.4 you alternate sums over an infinite horizon and sums over T time steps. You should stick to the T horizon case, as you mention the case T=1 later.

p6 Related work

The link to the work of Salimans 2017 is far from obvious, I would be glad to know more...

Q-prop (Haarnoja et al.,2017): this is not the adequate reference to Q-prop, it should be (Gu et al. 2016), you have it correct later ;)

Figure 3: why do you stop after so few epochs? I wondered how expensive is the computation of your estimator, but since in the RL case you go up to 50 millions (or 4 millions?), it's probably not the issue. I would be glad to see another horizontal lowest validation error for your RELAX estimator (so you need to run more epochs).
"ELBO" should be explained here (it is only explained in the appendices).

6.2, Table 1: Best obtained training objective: what does this mean? Should it be small or large? You need to explain better. How much is the modest improvement (rather give relative improvement in the text?)? To me, you should not defer Table 3 to an appendix (nor Table 4).

Figure 4: Any idea why A2C oscillates so much on inverted pendulum? Any idea why variance starts to decrease after 500 episodes using RELAX? Isn't related to the combination of regression and optimization, as suggested above?

About Double Inverted Pendulum, Appendix E3 mentions 50 million frames, but the figure shows 4 millions steps. Where is the truth?

Why do you give steps for the reward, and episodes for log-variance? The caption mentions "variance (log-scale)", but saying "log-variance" would be more adequate.

p9: the optimal control variate: what is this exactly? How do you compare a control variate over another? This may be explained in Section 2.

GAE (Kimura, 2000). I'm glad you refer to former work (there is a very annoying tendency those days to refer only to very recent papers from a small set of people who do not correctly refer themselves to previous work), but you may nevertheless refer to John Schulman's paper about GAEs anyways... ;)

Appendix E.1 could be reorganized, with a common hat and then E.1.1 for one layer model(s?) and E.1.2 for the two layer model(s?)

A sensitivity analysis wrt to your hyper-parameters would be welcome, this is true for all empirical studies.

In E2, is the output layer linear? You just say it is not ReLU...

The networks used in E2 are very small (a standard would be 300 and 400 neurons in hidden layers). Do you have a constraint on this?

"As our control variate does not have the same interpretation as the value function of A2C, it was not directly clear how to add reward bootstrapping and other variance reduction techniques common in RL into our model. We leave the task of incorporating these and other variance reduction techniques to future work."
First, this is important, so if this is true I would move this to the main text (not in appendix).
But also, it seems to me that the first sentence of E3 contradicts this, so where is the truth?

{0.01,0.003,0.001} I don't believe you just tried these values. Most probably, you played with other values before deciding to perform grid search on these, right?
The same for 25 in E3.

Globally, you experimental part is rather weak, we would expect a stronger methodology, more experiments also with more difficult benchmarks (half-cheetah and the whole gym zoo ;)), more detailed analyses of the results, but to me the value of your paper is more didactical and conceptual than experimental, which I really appreciate, so I will support your paper despite these weaknesses.

Good luck! :)

---------------------------------------
Typos:

p5
monte-carlo => Monte(-)Carlo (no - later...)
taylor => Taylor
you should always capitalize Section, equation, table, figure, appendix, ...

gradient decent => descent (twice)

p11: probabalistic

p15 ELU(Djork-... => missing space

---

> ### Author Response · Authors · 2017-12-31
> **Global Comments**
>
> Dear reviewer 1,
>
> Thank you for your detailed comments and positive reception of the work. I will address your comments in the order in which you wrote them. For clarity, I have placed your comments in [brackets] and my responses will follow in plain text.
>
> Global Comments:
>
> [When reading (6), it is clear that the framework performs regression of $c_\phi$ towards the unknown $f$ simultaneously with optimization over $c_\phi$.
> Taking this perspective, I would be glad to see how the regression part performs with respect to standard least square regression,
> i.e. just using $||f(b)-c_\phi(b)||^2$ as loss function. You may compare the speed of convergence of $c_\phi$ towards $f$ using (6) and the least squared error.
> You may also investigate the role of this regression part into the global g_LAX optimization by studying the evolution of the components of (6).]
>
> We did not experiment with the L2 loss directly. I do believe this should be explored in further work as the L2 loss has less computational overhead than the Monte Carlo variance estimate that was used in this work. This was tested in the concurrently submitted “Sample-efficient Policy Optimization with Stein Control Variate“ and in that work, the Monte Carlo variance estimate was found to produce better results in some settings. We ourselves are curious about the relative performance of minimizing the L2 loss vs the variance, but will probably leave that for further work.
>
> [Related to the above comment, in Algo. 1, you mention "f(.)" as given to the algo. Actually, the algo does not know f itself, otherwise it would not be blackbox optimization. So you may mean different things. In a batch setting, you may give a batch of [x,f(x) (,cost(x)?)] points to the algo. You more probably mean here that you have an "oracle" that, given some x, tells you f(x) on demand. But the way you are sampling x is not specified clearly.]
>
> You are correct in your understanding the algorithm. For clarity of notation, we have decided to keep the algorithm box as it is.
>
> [This becomes more striking when you move to reinforcement learning problems, which is my main interest. The RL algorithm itself is not much specified. Does it use a replay buffer (probably not)? Is it on-policy or off-policy (probably on-policy)? What about the exploration policy? I want to know more... Probably you just replace (10) with (11) in A2C, but this is not clearly specified.]
>
> We do not use a reply-buffer, our method is on-policy. We simply replace (10) with (11) as you mentioned. Since we mention the similarities between our algorithm and A2C, we have decided to not elaborate further. Exploration is encouraged due to entropy regularization which is commonly used in policy-gradient methods. A note has been added to the experimental details section in the Appendix to explain this.
>
> [In Section 4, can you explain why, in the RL case, you must introduce stochasticity to the inputs? Is this related to the exploration issue (see above)?]
>
> We were trying to explain that none of these methods can be used without modification to optimize parameters of a deterministic discrete function, since there would be no exploration.  In the RL case, stochasticity is introduced by using a stochastic policy and exploration is encouraged due to entropy regularization. For brevity and clarity, this paragraph has been removed.
>
> [Last sentence of conclusion: you are too allusive about the relationship between your learned control variate and the Q-function. I don't get it, and I want to know more…]
>
> Here we are simply mentioning the potential of training the control variate using off policy data as is done in the Q-prop paper. We believe further theoretical work must be done to better understand the relationship between the optimal control variate and the optimal Q-function, so we do not make any claims about this relationship.
>
> (continued in next comment)

---

> > ### Author Response · Authors · 2017-12-31
> > **continued response: Local comments part 1**
> >
> > Local Comments:
> >
> > [Backpropagation through the void: I don't understand why this title. I'm not a native english speaker, I'm probably missing a reference to something, I would be glad to get it.]
> >
> > The title alludes to the scope of the method.  Normally we refer to backprop “through” something.  If we use our method to estimate gradients through an unknown or non-existent computation graph, we could say we are backpropagating through “the void”.  However, I would guess that you are not the only one confused by the title.
> >
> > [Figure 1 right. Caption states variance, but it is log variance. Why does it oscillate so much with RELAX?]
> >
> > The caption has been changed to log-variance. We guess that the oscillation is due to interactions that arise as the control variate is trained. Since the parameters of the distribution p are constantly changing, the control variate is consistently a step behind p, which may account for these oscillations. We are aware of this phenomenon but have left it for further work to analyze in more detail.
> >
> > [Beginning of 3.1: you may state more clearly that optimizing $c_\phi$ the way you do it will also "minimize" the variance, and explain better why ("we require the gradient of the variance of our gradient estimator"...). It took me a while to get it.]
> >
> > We believe it is clear that by minimizing a Monte Carlo estimate of the variance, we will be minimizing the variance of the estimator. The first sentence of the second paragraph in section 3.1 has been slightly changed to avoid confusion. The title of the section has also been changed to be more clear.
> >
> >
> >
> > [In 3.1.1 a weighting based on $\d/\d\theta log p(b)$ => shouldn't you write $... log p(b|\theta)$ as before? ]
> >
> > The paper has been updated to include the dependence on theta in the distribution.
> >
> >
> > [Figure 2 is mentioned in p.3, it should appear much sooner than p6.]
> >
> > Due to length restrictions, it was difficult to place this figure elsewhere in the paper.
> >
> >
> >
> > [In Figure 2, there is nothing about the REINFORCE PART. Why?]
> >
> > We wanted to focus on comparing the learned control variates of REBAR and RELAX.
> >
> >
> > [In 3.4 you alternate sums over an infinite horizon and sums over T time steps. You should stick to the T horizon case, as you mention the case T=1 later.]
> >
> > Good point, we changed our notation to use a finite horizon of T.
> >
> >
> > [p6 Related work]
> >
> > Can you clarify what you mean here?
> >
> >
> > [The link to the work of Salimans 2017 is far from obvious, I would be glad to know more…]
> >
> > When applied to RL, our algorithm is an extension of policy-gradient optimization. We felt it reasonable to refer to alternative approaches such as that of Salimans et al. to inform the reader of concurrent, but orthogonal work.
> >
> >
> >
> > [Q-prop (Haarnoja et al.,2017): this is not the adequate reference to Q-prop, it should be (Gu et al. 2016), you have it correct later ;)]
> >
> > Thanks, we corrected this.
> >
> > [Figure 3: why do you stop after so few epochs? I wondered how expensive is the computation of your estimator, but since in the RL case you go up to 50 millions (or 4 millions?), it's probably not the issue. I would be glad to see another horizontal lowest validation error for your RELAX estimator (so you need to run more epochs).
> > "ELBO" should be explained here (it is only explained in the appendices).]
> >
> > In our VAE experiments, we were interested in comparing with the baseline of REBAR. For that reason, we followed their experimental setup which, runs for 2 million iterations.  We added a note that ELBO is shorthand for evidence lower-bound.
> >
> >
> > [6.2, Table 1: Best obtained training objective: what does this mean? Should it be small or large? You need to explain better. How much is the modest improvement (rather give relative improvement in the text?)? To me, you should not defer Table 3 to an appendix (nor Table 4).]
> >
> > Good points.  We change “best objective”to “highest obtained ELBO” to indicate that higher ELBOs are better. We couldn’t move the tables up because of space constraints.
> >
> > [Figure 4: Any idea why A2C oscillates so much on inverted pendulum? Any idea why variance starts to decrease after 500 episodes using RELAX? Isn't related to the combination of regression and optimization, as suggested above?]
> >
> > Regarding the inverted pendulum, we believe this is due to the larger variance of the baseline gradient estimator. Regarding the variance on the double pendulum, we do not fully understand this phenomenon, thus we do not make any claims about it.
> >
> > (continued in next comment)

---

> > > ### Author Response · Authors · 2017-12-31
> > > **Local comments part 2**
> > >
> > > [About Double Inverted Pendulum, Appendix E3 mentions 50 million frames, but the figure shows 4 millions steps. Where is the truth?]
> > >
> > > This was due to a typo in the appendix. The figure is correct- it was run for 5 million steps, and we corrected the appendix to reflect this.
> > >
> > >
> > >
> > > [Why do you give steps for the reward, and episodes for log-variance? The caption mentions "variance (log-scale)", but saying "log-variance" would be more adequate.]
> > >
> > > The variance was estimated at every episode since we needed to run a full episode to compute the policy gradient. After every training episode, 100 episodes were run to generate samples from our gradient estimator which were used to estimate the variance of the estimator. We run the algorithms for a fixed number of steps to be consistent with previous work.
> > >
> > >
> > >
> > > [p9: the optimal control variate: what is this exactly? How do you compare a control variate over another? This may be explained in Section 2.]
> > >
> > > The optimal control variate is the control variate which produces a gradient estimator with the lowest possible variance.
> > >
> > >
> > >
> > > [GAE (Kimura, 2000). I'm glad you refer to former work (there is a very annoying tendency those days to refer only to very recent papers from a small set of people who do not correctly refer themselves to previous work), but you may nevertheless refer to John Schulman's paper about GAEs anyways... ;)]
> > >
> > > Thanks for the pointer, this reference was added.
> > >
> > >
> > > [Appendix E.1 could be reorganized, with a common hat and then E.1.1 for one layer model(s?) and E.1.2 for the two layer model(s?)]
> > >
> > > The appendix was reorganized as per your suggestions.
> > >
> > >
> > >
> > > [A sensitivity analysis wrt to your hyper-parameters would be welcome, this is true for all empirical studies.]
> > >
> > > In general, we did not find the algorithm to be very sensitive to hyperparameters other than the learning rate.  We agree that adding a sensitivity analysis would be an improvement. We have added this to the experimental details in the Appendix along with a sentence which presents the best performing hyperparameters.
> > >
> > >
> > > [In E2, is the output layer linear? You just say it is not ReLU…]
> > >
> > > The Appendix was changed to note that the output layers were linear.
> > >
> > >
> > > [The networks used in E2 are very small (a standard would be 300 and 400 neurons in hidden layers). Do you have a constraint on this?]
> > >
> > > There was no hard constraint on network size. These networks were chosen because they worked well with the baseline A2C algorithm, and is a standard choice in the literature, and the OpenAI baselines.
> > >
> > >
> > > ["As our control variate does not have the same interpretation as the value function of A2C, it was not directly clear how to add reward bootstrapping and other variance reduction techniques common in RL into our model. We leave the task of incorporating these and other variance reduction techniques to future work."
> > > First, this is important, so if this is true I would move this to the main text (not in appendix).
> > > But also, it seems to me that the first sentence of E3 contradicts this, so where is the truth?]
> > >
> > >
> > > We moved this section to the main text, and clarified why we don’t use reward bootstrapping for discrete, but use it for continuous experiments. We do so with the continuous control RL tasks by structuring the control variate as C = V(s) + c(a, s), where V was trained as the value function in A2C.
> > >
> > >
> > > [{0.01,0.003,0.001} I don't believe you just tried these values. Most probably, you played with other values before deciding to perform grid search on these, right?
> > > The same for 25 in E3.]
> > >
> > > For all the experiments, the hyperparameter values used for grid search mentioned in the appendices (E1, E2, E3) were the only ones tried. No other values were tried because of computational constraints. These values were not chosen because they gave us better result, but because we thought they would make a comprehensive grid.
> > >
> > >
> > > [Globally, you experimental part is rather weak, we would expect a stronger methodology, more experiments also with more difficult benchmarks (half-cheetah and the whole gym zoo ;)), more detailed analyses of the results, but to me the value of your paper is more didactical and conceptual than experimental, which I really appreciate, so I will support your paper despite these weaknesses.]
> > >
> > > Yes, we believe that further experimentation is warranted, but feel that our current results demonstrate the effectiveness of our method. Thank you for your support.
> > >
> > > Typos:
> > >
> > > We have fixed all of the typos that you noticed. Thanks.

---

### Official Review · AnonReviewer3 · 2017-11-27
**I think the paper provides interesting idea for optimizing black-box functions of random variables.**

**Rating:** 6
**Confidence:** 2

**Review:**

The paper considers the problem of choosing the parameters of distribution to maximize an expectation over that distribution. This setting has attracted a huge interest in ML communities (that is related to learning policy in RL as well as variational inference with hidden variables). The paper provides a framework for such optimization, by interestingly combining three standard ways.

Given Tucker et al, its contribution is somehow incremental, but I think it is an interesting idea to use neural networks for control variate to handle the case where f is unknown.

The main issue of this paper seems to be in limited experimental results; they only showed three quite simple experiments (I guess they need to focus one setting; RL or VAE). Moreover, it would be good to actually show if the variation of g_hat is much smaller than other standard methods.

There is missing reference at page 8.

---

> ### Author Response · Authors · 2017-12-31
> **Response to reviewer 3**
>
> Dear Reviewer 3,
>
> Thank you for your overall positive review of the work.  To respond to your specific criticisms:
>
> 1) “Given Tucker et al, its contribution is somehow incremental”.  We did build closely on Tucker et. al., but we generalized their method, and expanded its scope substantially.  REBAR was only applicable to known computation graphs with discrete random variables, making it inapplicable to reinforcement learning or continuous random variables.
>
> 2) “ they only showed three quite simple experiments (I guess they need to focus one setting; RL or VAE)”.  Our first experiment was deliberately simple, so we could visualize the learned control variate.  Regarding our VAE experiments, achieving state of the art on discrete variational autoencoders may constitute a “simple” experiment, but it’s not clear that this is a problem.  We included the RL experiments to demonstrate the breadth of problems to which our method can be applied, since this is one of the main advantages of RELAX over REBAR.
>
> 3) “it would be good to actually show if the variation of g_hat is much smaller than other standard methods”.  Figures 1 and 4 both have plots labeled ‘log variance’, showing exactly this.

---

### Public Comment · ~Leon_Bottou1 · 2017-11-27
**Relation with doubly robust estimation**

First of all, this is a very nicely written paper on an important topic.

I only write this comment because I would like to hear the comments of the author(s) about the following remarks.

    1)  To me equation (7) expresses the doubly robust estimation of the gradient in question. This is a bit different from the usual robust estimation because the importance sampling weights, dlog(P(b,theta))/dtheta are signed, but this remains fundamentally similar.

    2)  One very interesting aspect of the paper might therefore be a clarification of the power of the reparametrization trick.  It now looks like doubly robust estimation of the gradient using the knowledge that the predictor function is exact (that makes it simply robust in fact ;-)  When it is not exact, one needs to correct with a pair of weighted estimates.

    3) How much of the power of the reparametrization trick comes from the fact that the reparametrization is assumed exact, as opposed to needing to be corrected by a difference of weighted estimates?

Many thanks.

---

> ### Author Response · Authors · 2018-01-03
> **That's a useful connection**
>
> Dear Leon, you make some interesting connections!
>
> 1) Yes, our estimator does look a lot like a doubly-robust estimator.  In fact, estimating gradients through discrete random variables seems a lot like counterfactual modeling, where we happen to know all the confounders.  The motivation seems a bit different, though, in that the doubly-robust estimator is about correcting for model misspecification and removing bias, where we start with an unbiased estimator.  But the overall idea is indeed similar.  Also, the fact that these different weighting schemes both give unbiased estimates raises the question: what is the family of weighting schemes that we could be looking at?
>
> 2) I see what you mean - the special case where c = f recovers the reparameterization trick, and the weighted estimates cancel out.  One thread I'd like to think about further is, can we do even better?  We actually have results on toy problems where LAX achieves lower variance than the reparameterization trick.  Of course, there are sometimes many possible reparameterizations, each with different variance.
>
> 3) I think the "Generalized Reparameterization Gradient" paper looked at questions very similar to this one.  Also, one future direction we're thinking about is learning the reparameterization as well as the surrogate.
>
>
> Thanks again for the interesting points.

---

### Public Comment · (anonymous) · 2017-12-12
**Missing related work?**

Other recent work on generalizing reparameterization gradients seems to be missing:
Ruiz et al., The Generalized Reparameterization Gradient, NIPS, 2016.
Naesseth et al., Reparameterization Gradients through Acceptance-Rejection Sampling Algorithms, AISTATS, 2017.

---

> ### Author Response · Authors · 2018-01-03
> **thanks!**
>
> Thank you for your comment. A sentence has been added to section 7 to mention the application these works could have to our method.

---

### Decision · Program_Chairs · 2018-01-29
**ICLR 2018 Conference Acceptance Decision**

**Decision:**

Accept (Poster)

**Comment:**

This is an interesting and well-written paper introducing two unbiased gradient estimators for optimizing expectations of black box functions. LAX can handle functions of both continuous and discrete random variables, while RELAX is specialized to functions of discrete variables and can be seen as a version of the recently introduced REBAR with its concrete-relaxation-based control variate replaced by (or augmented with) a free-form function. The experimental section of the paper is adequate but not particularly strong. If Q-prop is the most similar existing RL approach, as is state in the paper, why not include it as a baseline? It would also be good to see how RELAX performs at optimizing discrete VAEs using just the free-form control variate (instead of combining it with the REBAR control variate).